# Contraceptive use among sexually active women living with HIV in western Ethiopia

**Tesfaye Regassa Feyissa** [1,2] *, **Melissa L. Harris** [2], **Peta M. Forder** [2], **Deborah Loxton** [2]

**1** College of Health Science, Wollega University, Nekemte, Oromia, Ethiopia, **2** Research Centre for Generational Health and Ageing, School of Medicine and Public Health, Faculty of Health and Medicine, University of Newcastle, Callaghan, New South Wales, Australia

\* regassatesfaye@gmail.com

## Abstract

### Introduction

Contraception can help to meet family planning goals for women living with HIV (WLHIV) as well as to support the prevention of mother to child transmission of HIV (PMTCT). However, there is little research into the contraceptive practice among sexually active WLHIV in Ethiopia. Therefore, we aimed to examine contraceptive practice among sexually active WLHIV in western Ethiopia and identify the factors that influenced such practice using the Health Belief Model (HBM).

### Methods

A facility-based cross-sectional survey of 360 sexually active WLHIV was conducted from 19th March to 22nd June 2018 in western Ethiopia. The eligible participants were WLHIV aged between 18 and 49 years who reported being fecund and sexually active within the previous six months but were not pregnant and not wanting to have another child within two years. Modified Poisson regression analyses were conducted to identify factors that influenced contraceptive practice among sexually active WLHIV in western Ethiopia.

### Results

Among sexually active WLHIV (n = 360), 75% used contraception with 25% having unmet needs. Of the contraceptive users, 44.8% used injectables, 37.4% used condoms and 28.5% used implants. Among 152 recorded births in the last five years, 17.8% were reported as mistimed and 25.7% as unwanted. Compared to WLHIV having no child after HIV diagnosis, having two or more children after HIV diagnosis (Adjusted Prevalence Ratio [APR] = 1.31; 95%CI 1.09–1.58) was associated with increased risk of contraceptive practice. However, sexually active unmarried WLHIV (APR = 0.69; 95%CI 0.50–0.95) were less likely to use any contraception compared to their sexually active married counterparts. Importantly, high perceived susceptibility (APR = 1.49; 95%CI 1.20–1.86) and medium perceived susceptibility (APR = 1.55; 95%CI 1.28–1.87) towards unintended pregnancy were associated with higher risk of contraceptive use than WLHIV with low perceived susceptibility.

**Data Availability Statement:** Due to the presence of potentially sensitive information provided by women living with HIV, data have been made available upon request. This requirement was imposed by Human Research Ethics Committee of

The University of Newcastle, Australia, and the Oromia Regional State Health Bureau Research Ethics Committee, Ethiopia which approved the research protocol. The data request may be submitted to the Research Centre for Generational Health and Ageing, University of Newcastle, Australia at rcgha@newcastle.edu.au.

**Funding:** This study was partially supported by the Hunter Medical Research Institute/Greaves Family Postgraduate Top-Up Scholarship (Grant number G1701582). Wollega University (first author's employer organization) facilitated the data collection process. TRF is supported by The University of Newcastle International Postgraduate Research Scholarship (UNIPRS) and The University of Newcastle Research Scholarship Central 50:50 (UNRSC 50:50). Dr Melissa Harris is supported by an Australian Research Council Discovery Early Career Researcher Award (DECRA). The funders had no role in study design, data collection and analysis, decision to publish, or preparation of the manuscript.

**Competing interests:** The authors have declared that no competing interests exist.

**Abbreviations:** AIDS, Acquired Immunodeficiency Syndrome; APR, Adjusted Prevalence Ratio; ART, Antiretroviral therapy; CI, Confidence interval; HBM, Health Belief Model; HIV, Human Immunodeficiency Virus; HREC, Human Research Ethics Committee; IUD, Intrauterine device; KMO, Kaiser-Meyer-Olkin; PCA, Principal component analysis; PMTCT, Prevention of mother to child transmission of HIV; PR, Prevalence Ratio; REDCap, Research Electronic Data Capture; SD, Standard deviation; SSA, Sub-Saharan Africa; STIs, Sexually transmitted infections; WHO, World Health Organization; WLHIV, Women living with HIV.

## Conclusions

Although contraceptive use amongst sexually active WLHIV was found to be high, our findings highlight the need for strengthening family planning services given the high rate of unintended pregnancies, the high rate of unmet needs for contraception, as well as the lower efficacy with some of the methods. Our findings also suggest that the HBM would be a valuable framework for healthcare providers, programme planners and policymakers to develop guidelines and policies for contraceptive counselling and choices.

## Introduction

In 2017, nearly two-thirds of the 25.7 million people living with human immunodeficiency virus (HIV) in sub-Saharan Africa (SSA) were women [1, 2], with similar proportions identified in Ethiopia [3]. The 2015 World Health Organization (WHO) guideline states there should be no criterion barrier in the initiation of antiretroviral therapy (ART) [4] which helped with reductions in HIV-related morbidity and mortality [5]. With the current ART expansion, meeting contraceptive needs is crucial to not only achieving fertility goals [6] but also reductions in maternal mortality [7–9], child mortality and children who are being orphaned [9]. Contraception could also play a critical role in the prevention of mother to child transmission of HIV (PMTCT) [10] because it averts unintended births [11]. Therefore, strengthening contraceptive programs could play a role in ending the epidemics of acquired immunodeficiency syndrome (AIDS) by 2030 (the Sustainable Development Goal 3.3) [12].

Despite the benefits of contraception, there are considerable contraceptive use gaps amongst women living with HIV (WLHIV) [13, 14], particularly in SSA. With high rates of unintended pregnancy [11] and abortion [15], there appears to be a high unmet need for contraception among WLHIV [16] and contraceptive failure [17]. According to the WHO, women with an unmet need are "those who are fecund and sexually active but are not using any method of contraception, and report not wanting any more children or wanting to delay the next child" [18].

With improved survival of the HIV-positive population because of ART [19, 20], WLHIV need clear reproductive life plans with ready access to contraception. Studies that have examined contraceptive practice among WLHIV have focused on married women [20, 21]. This places unmarried women who are sexually active at increased risk of unintended pregnancy due to the assumption made about sex only happening within the context of marriage (and therefore a lack of reproductive planning for these women through health services) [22].

While it has been advised by the WHO that WLHIV have the right to choose any contraceptive methods similar to HIV-negative women [17], the contraceptive choice in the presence of HIV appears more complex because WLHIV are required to balance the prevention of both unintended pregnancy and HIV transmission [23, 24].

Evidence suggests that theory-based research can explain not only contraceptive behaviors but also provide strategies for increased uptake and continuation [25]. In contraceptive research, the Health Belief Model (HBM) offers an important perspective whereby *modifying and enabling factors* (such as socio-demographic, structural, psychological and reproductive factors) interplay with personal perceptions and cues to action to influence the decision to use

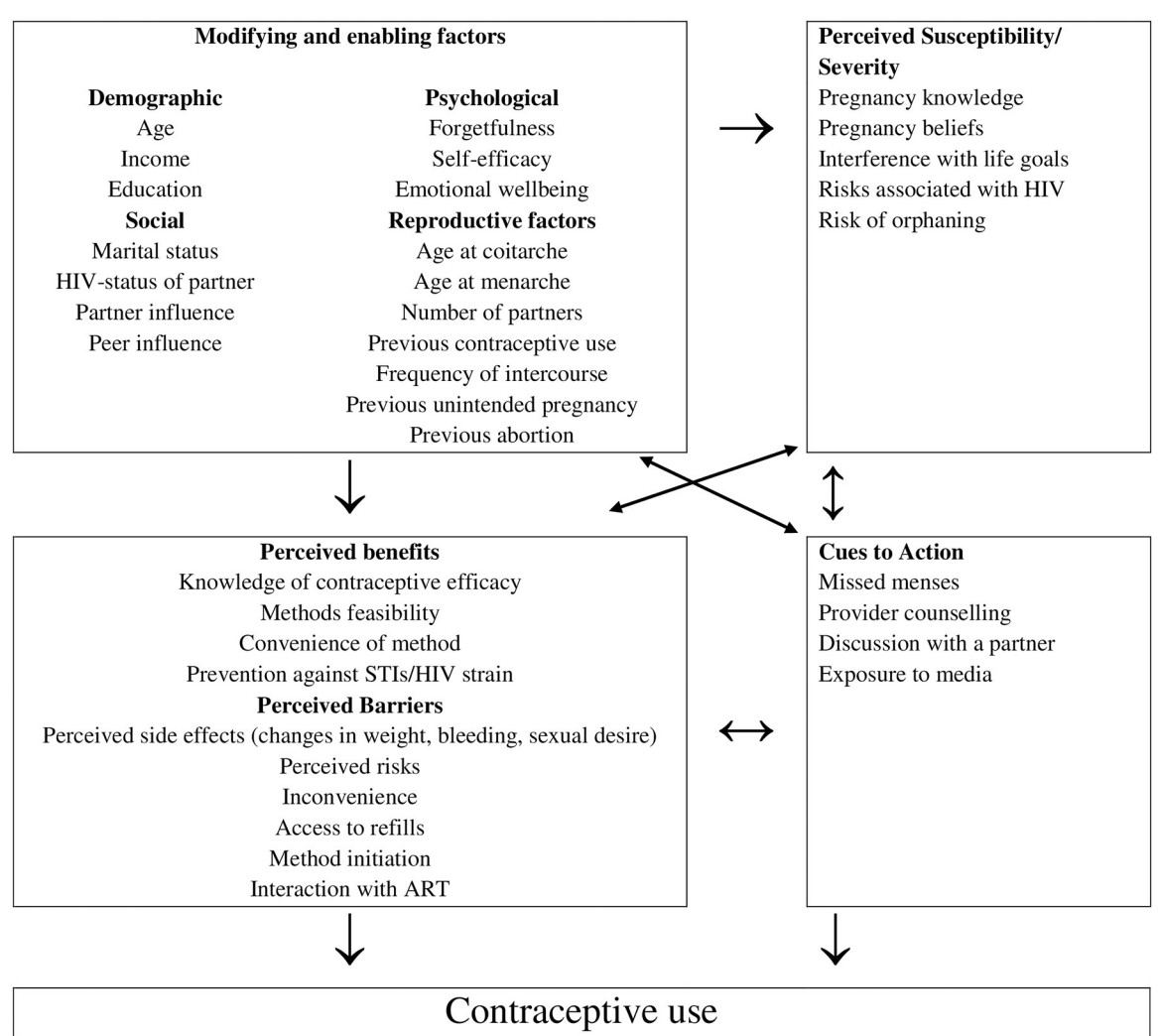

**Fig 1. Contraceptive behaviour of women living with HIV using the health belief model.**

contraceptives [26] (see Fig 1). According to the HBM, there are four constructs of perceptions: (a) *perceived susceptibility* to dictate the perceived chance of pregnancy (unintended) if contraception is not used; (b) a *perceived severity* that describes consequences of getting and being pregnant amongst WLHIV which include perceptions regarding mother to child transmission of HIV; (c) *perceived benefits* of contraception which include effectiveness, dual protection, and reversibility of the methods; and (d) *perceived barriers* of contraceptive use include side effects of contraception. *Cues to action* are internal and external signals used to activate readiness to change [26]. There are several studies that have used the HBM to predict contraceptive utilization [27–31]; however, no study has used the HBM in relation to contraceptive use amongst WLHIV.

To inform policy and programs regarding the contraceptive use for sexually active WLHIV, understanding WLHIV's contraceptive use within this framework is crucial [32]. Therefore, the present study aimed to evaluate contraceptive use among sexually active WLHIV in western Ethiopia using the HBM as a conceptual framework.

## Materials and methods

### Study design and settings

This study used a facility-based cross-sectional survey among WLHIV in western Ethiopia. The study was conducted in East and West Wollega Zones of western Ethiopia from 19[th] March to 22[nd] June 2018.

### Study population and sampling

The study participants were recruited using a systematic sampling of HIV-positive women attending HIV clinics across four health facilities (two hospitals and two health centers). The recent treat-all policy meant there was no criteria barrier to starting ART [4], encouraging all PLHIV to commence ART immediately after HIV-positive diagnosis. To facilitate this in Ethiopia, a unique ART number is assigned to PLHIV to monitor their ART treatment and HIV management, and to coordinate care across different settings. For the recruitment process, a list of ART clients (each with a unique ART number) was obtained from the daily appointment calendar, excluding the personal identifiers of each client. Among 2,445 reproductive-age WLHIV, a systematic sampling of every second woman from each health facility was used to select study participants with selected WLHIV invited to participate in the study. All participants were contacted at a HIV clinic of the selected health institutions when they came for their ART services.

Eligibility for this analysis was restricted to non-pregnant WLHIV who reported: (a) being sexually active within the last six months at survey completion; (b) being fecund; (c) not wanting to have another child within two years [33, 34]; and (d) completed the questions on contraceptive use. As shown in Fig 2, among the 1,082 women who were surveyed, 360 sexually active WLHIV who met the eligibility criteria were included for analysis.

### Data collection procedures

Data collection was conducted using a standard survey questionnaire which was developed based on the Ethiopian Demographic and Health Survey questionnaire [35] and guided by existing literature on contraception, as well as concepts within the HBM [26]. The tool was originally prepared in English and subsequently translated into the local language, Oromo. Data quality was maintained by training data collectors on the questionnaire, consenting procedures and completing surveys. To ensure validity (measuring what it aims to measure) and reliability (consistency of measurements) [36] of the questionnaire, the questions were mainly adapted from the Demographic and Health Survey (https://dhsprogram.com/pubs/pdf/FR328/FR328.pdf) [35] and Family Planning 2020 questionnaires (https://www.familyplanning2020.org/) [34], which were widely tested in many countries, including Ethiopia. Because of ethical and logistic reasons, a pilot study was carried out on 30 participants in the health facilities that were selected for main data collection. These women were not included the main data collection. Based on those results and feedbacks, some questions were modified (such as contraceptive use and fertility characteristics) for the main survey. The training and pilot testing were conducted for five days. Data were collected by five trained female nurses who had previous experience in data collection, and were fluent in the local language. The data collectors were never involved in the participants' care. Data collection was conducted face-to-face with mobile-based surveys. The Research Electronic Data Capture (REDCap) software [37] was used for data collection. Data collection was overseen by a supervisor for completeness and consistency. Furthermore, the data completeness, accuracy and consistency across the data

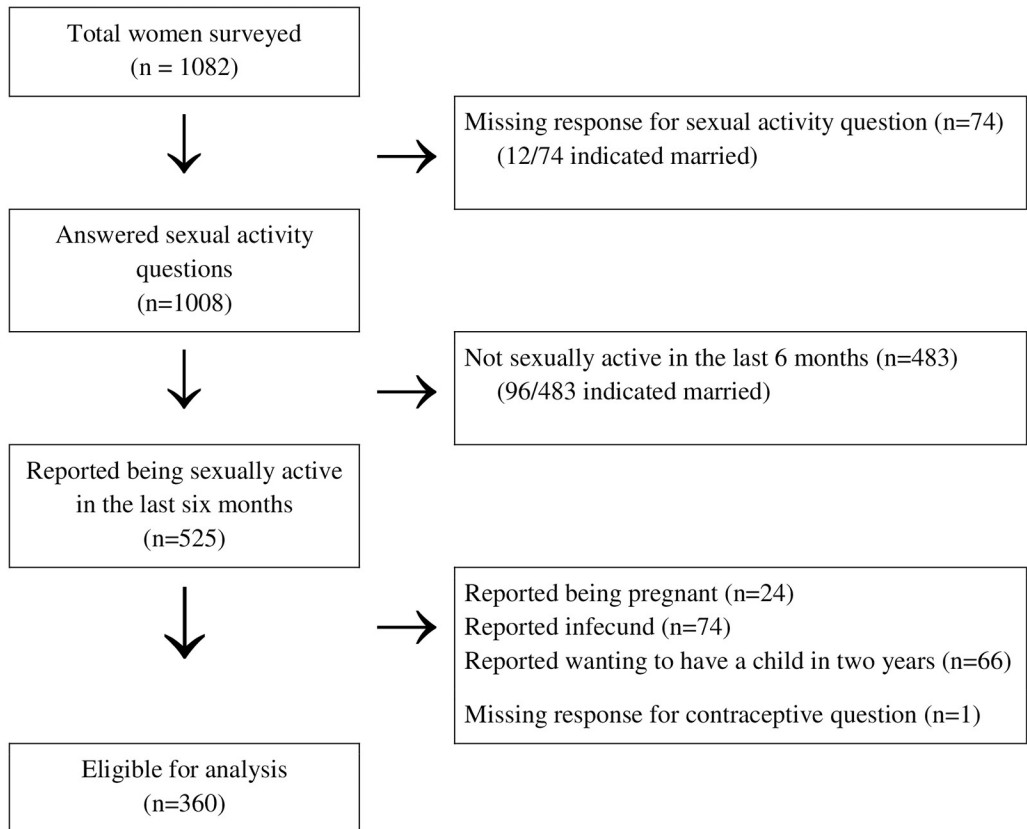

**Fig 2. Flow chart illustrating the eligibility process to obtain the final sample for analysis of sexually active WLHIV.**

collectors were checked in the REDcap database after transferring data to the database daily. Any concerns were further discussed during the next morning with data collectors.

## Ethical considerations

Ethical approval for this study was obtained from The Human Research Ethics Committee (HREC) of The University of Newcastle, Australia (H-2017-0289), and the Oromia Regional State Health Bureau Research Ethics Committee, Ethiopia (BEFO/HBISH/1-16/257). Official permission was obtained from hospitals, health centers, and respective HIV-clinics included in our study. An information statement was provided to all participants prior to obtaining informed verbal consent. To ensure informed verbal consent from participants, the data collectors read the information statement in the local language, Oromo. There are low literacy levels among women in Ethiopia, (42% of women in Ethiopia are literate) [35], so informed verbal consent was more appropriate and was approved by both ethics committees. In addition, it was a survey and the research involved no more than low risk [38]. Participants were given the opportunity to ask questions prior to the interview. The consent procedure took place in a separate private room by female nurse data collectors after WLHIV had finished their routine clinical care appointment. Participants were informed that their participation was voluntary and that they were free to decline participation or withdraw their consent at any time. It was made clear that participation in this study had no bearing on their receipt of clinical care. The participants were also informed that the survey involved some questions that they might find embarrassing or too personal and some that might cause them to worry about their

reproductive health issues. Further, participants were informed that they did not have to answer any question that they did not feel comfortable with, and they could withdraw at any time or simply choose not to answer a particular question. Female nurses were prepared to provide psychological support if the need arose. Anonymised data were stored on password-protected Ipads during data collection. Data were stored on secure and password-protected computers.

## Measures

**Outcome variables (contraceptive use).**   Participants were asked if they had done something or used any method to delay or avoid getting pregnant during the data collection period, i.e., between March and June 2018 (yes/no). Contraceptive users were asked about the method (s) used, which included: short-acting contraception (pills, condoms and injectables); long-acting reversible contraception (implants and Intra-Uterine Devices [IUDs]); and permanent contraception (vasectomy and tubal ligation). Simultaneous use of condoms and any other method(s) was defined as dual use for this study. Based on responses, we further created five exclusive groups of contraceptive users: (a) condom use only; (b) short-acting contraception only (pills, injectables); (c) dual use of condoms plus short-acting contraception (pills, injectables); (d) long-acting only (IUDs, implants); and (e) dual use of condom and long-acting contraception (IUDs, implants). Contraceptive continuation was also determined by the length of time the women were using the method(s) without interruption ($\leq$12 months, >12 to < 36 months, and $\geq$ 36 months). Contraceptive users were also surveyed with respect to the following: partners' support of their contraceptive use (supportive, indifferent and not supportive); the source of the recent methods (hospital, health center, health post, private-for-profit, and non-government organization); whether there was counselling on possible side effects of contraceptives (yes/no); and whether they were informed about what to do concerning side effects (yes/no). Those women who reported not using contraception were also asked if they had intended to use contraception (yes/no).

**Explanatory factors.**   Using the HBM framework, potential predictors of contraceptive use focused on (a) modifying and enabling factors; (b) perceptions regarding conception and contraception; and (c) cues to action. The dimensions of modifying and enabling factors included a broad range of the following: (i) socio-demographic characteristics, (ii) HIV-related factors, and (iii) reproductive characteristics.

*Socio-demographic characteristics included.* the type of health facility being accessed for ART (categorized as hospitals and health centers); age (in years); marital status (categorized as married and unmarried); residence (urban and rural); schooling (no formal education, primary education, secondary education, and any post-secondary education); monthly family income (less than 1500 Ethiopian Birr, $\geq$ 1500 Ethiopian Birr, and don't know); and main decision-maker regarding income use (respondent, partner or joint decisions made) [35]. Furthermore, travel time to health facility (<60 minutes, 60 to <120 minutes, and $\geq$120 minutes) and round cost of travel (<25 Birr, 25 to <50 Birr, and $\geq$50 Birr) were also assessed.

*HIV-related factors included.* time since HIV diagnosis and time on ART in years, which were grouped into three ($\leq$5 years, 5 to <10 years, and $\geq$10 years); reported health status after ART initiation (poor/quite poor, neither good nor poor, quite good, and very good); partner tested for HIV (yes, no and don't know); and HIV status of tested partner (HIV-negative, HIV-positive and don't know). The recent CD4 count was collapsed into two groups based on previous thresholds for initiating ART (<350 and $\geq$350 cells/$\mu$L) [39].

*Reproductive characteristics included.* number of children living in the household (no children, 1–2 children, 3 or more); number of children not living with mother at home (no

children, 1 child, 2 or more); and number of children born since HIV diagnosis (no children, 1 child, 2 or more). Furthermore, pregnancy intention at conception of all live births in the last five years of these women was categorized into intended (*wanted then*), mistimed (*wanted later*) and unwanted (*not at all*). HIV status of mothers at conception in the last five years was also measured (HIV-negative, HIV-positive and not known).

Personal perceptions related to conception and contraception amongst WLHIV was measured using 30 questions as guided by existing HBM research on contraceptive use [28, 40]. Perceptions regarding (a) conception susceptibility (2 items), (b) severity (6 items), (c) contraception benefits (7 items), and (d) contraception barriers (11 items) were measured using five-point Likert scales (e.g., very unlikely to very likely, lowest to highest importance, strongly disagree to strongly agree, or similar). *Conception susceptibility* was assessed using questions about the perceived likelihood of becoming pregnant if contraception was not used and concerns of perceived difficulty if found pregnant right now (at the time of data collection). Perceptions regarding *severity* included questions on: HIV transmission to partner/respondent; HIV transmission during pregnancy; HIV transmission during childbirth; HIV transmission during breastfeeding; fear of orphaning the child; and stress if decided to have a baby. Perceptions regarding *contraceptive benefits* included questions on: long-term protection; convenience of methods; effectiveness; dual protection; ability to use discretely; reversibility; and the immediate return of menses after stopping the contraception. Perceptions regarding *contraceptive barriers* included questions on: pain related to implants/IUDs; inconvenience; frequency of visits to obtain the method; interaction with ART drugs; partner influence; weight gain; sexual interruptions; increased menstrual bleeding; cessation of menstrual bleeding; and bleeding between menses. Interaction with ART drugs; sexual interruptions; cessation of menstrual bleeding; bleeding between menses; frequency of visits to obtain the method; increased menstrual bleeding; and weight gain were recoded (from 1 to 5) for consistency across all other variables.

*The cues to action construct included four items measuring*: provider counselling; discussion with a partner about contraceptive options; media exposure to contraceptive information; and missed menses to start contraception after intercourse.

## Statistical analyses

The perception constructs and the cues to action construct (according to the HBM) were included in a principal component analysis (PCA) to summarize the data using Kaiser's varimax rotation method and a fixed number of factors. Eigenvalues were checked, with values greater than one essential to summarize using PCA. Sampling adequacy was then tested using Kaiser-Meyer-Olkin (KMO; with the threshold of 0.5 set for adequacy). Each of the five HBM PCA summed scores (conception susceptibility, severity, contraceptive benefits, contraceptive barriers and cues to action) were subsequently categorized into tertiles (e.g., high susceptibility, medium susceptibility and low susceptibility) [41].

Differences between contraceptive users and non-users were described with respect to the modifying and enabling factors as well as perceptions. Observed differences were evaluated using the Pearson chi-square (for categorical variables) and independent t-tests (for continuous variables) where applicable (level of significance at p <0.05). Fisher's exact test was used where the Pearson chi-square was not appropriate. Following this, we initially used log-binomial regression models to obtain the prevalence ratio (PR) [42, 43], although convergence was not achieved, even when using the COPY adjustment method. As a result, Poisson regression models with robust standard errors were used to estimate prevalence ratios [42, 43]. Poisson regression models were initially used to identify potential factors associated with contraceptive

use. Variables that were considered potential independent risk factors from the univariate analyses (p-value<0.2) were considered for a final multivariable model, to control for confounding. The strength of the association between the outcome variable and independent variables was expressed in PR with a 95% confidence interval (CI). All analyses were conducted using STATA®, version 14 (Stata Corporation, College Station, TX, USA).

## Results

### Socio-demographic characteristics

Among 360 sexually active WLHIV who were included in this study, contraceptive use was reported by 270 (75.0%) participants with a total unmet need of 25.0%. The mean age of the study participants was 31.7 years. Two hundred and ninety one (80.8%) participants were married. The majority of the participants (60.0%) had a follow-up in hospitals for ART. The proportion of women who reported a monthly family income of less than 1,500 Ethiopian Birr was 52.2%. Regarding the decision-making on the utilization of their income, 36.7% of participants reported that they could decide on income jointly with their partner. There were significant differences between contraceptive users and non-users with respect to health facility, age, marital status, monthly family income, and decision-making regarding income use (see Table 1).

### HIV-related, reproductive-related and HBM factors of sexually active women living with HIV

Results in Table 2 show the HIV-related, reproductive-related and HBM factors, with 111 (30.9%) women reporting that they learned of their HIV status 10 or more years ago. While all of the women were using ART (results not shown), 21.0% of them had been using ART for ten or more years. Overall, 77.2% had a CD4 cell count of greater than or equal to 350 with a mean of 611 cells/μL. Furthermore, 79.2% of women knew that their partner had been tested for HIV, of whom, 33.3% reported that their partner was HIV-negative. Only 14.5% did not have a child at home. Half of the women (48.5%) reported not having given birth since their HIV diagnosis. Among 152 recorded births given by the participants in the last five years, 17.8% were reported as mistimed births and 25.7% were reported as unwanted births. The conception of 129 (84.9%) births given by the participants were after learning of their HIV-positive result. The percentage of women who reported high perceived susceptibility to pregnancy (unintended) and severity of being pregnant while living with HIV were 20.8% and 30.7%, respectively. High perceived barriers and high perceived benefits were reported by 32.2% and 33.3% of participants, respectively. Finally, 33.3% reported high cues to action. There were significant differences between contraceptive users and non-users with respect to partner testing status, number of children living at home, number of children since HIV diagnosis, perceived susceptibility, perceived severity, perceived benefits and perceived barriers.

### Contraceptive use characteristics among sexually active women living with HIV

Among sexually active WLHIV, 75% used contraception. Of the women who reported contraceptive use, injectables were the most commonly used method (44.8%), followed by condoms (37.4%) and implants (28.5%). For a better understanding of contraceptive use, mutually exclusive categories were created for short-acting and reversible contraceptive users; 15.0% used condoms only, 33.5% used short-acting only, and 28.6% used long-acting only. Dual users of condoms with short-acting contraception (pills or injectables) were 15.0% while dual

**Table 1. Characteristics of sexually active women living with HIV in western Ethiopia (n = 360), according to use of contraceptives, 2018.**

| Characteristics | Categories | All women (N = 360) | | Contraceptive users (N = 270[75.0%]) | | Non-users of contraceptives (N = 90 [25.0%]) | | |
|---|---|---|---|---|---|---|---|---|
| | | n | % | n | % | n | % | p‡ |
| Health facility | Hospitals | 216 | 60.0 | 142 | 52.6 | 74 | 82.2 | <0.001 |
| | Health centers | 144 | 40.0 | 128 | 47.4 | 16 | 17.8 | |
| Age (years)† | | 31.7 (6.2) | | 31.2 (5.7) | | 33.1 (7.4) | | 0.011 |
| Marital status | Married | 291 | 80.8 | 239 | 88.5 | 52 | 57.8 | <0.001 |
| | Unmarried | 69 | 19.2 | 31 | 11.5 | 38 | 42.2 | |
| Residence | Urban | 338 | 93.9 | 256 | 94.8 | 82 | 91.1 | 0.20 |
| | Rural | 22 | 6.1 | 14 | 5.2 | 8 | 8.9 | |
| Schooling | No formal education | 76 | 21.2 | 61 | 22.6 | 15 | 17.1 | 0.67 |
| | Primary education | 186 | 52.0 | 136 | 50.4 | 50 | 56.8 | |
| | Secondary Education | 69 | 19.3 | 52 | 19.3 | 17 | 19.3 | |
| | Any post-secondary education | 27 | 7.5 | 21 | 7.8 | 6 | 6.8 | |
| | Missing | 2 | | 0 | | 2 | | |
| Monthly family income | Less than 1,500 Ethiopian Birr | 188 | 52.2 | 135 | 50.0 | 53 | 58.9 | 0.021 |
| | ≥1,500 Ethiopian Birr | 160 | 44.4 | 129 | 47.8 | 31 | 34.4 | |
| | Don't know | 12 | 3.3 | 6 | 2.2 | 6 | 6.7 | |
| Decision-regarding income | Respondent | 141 | 39.2 | 85 | 31.5 | 56 | 62.2 | <0.001 |
| | Husband/partner/family | 87 | 24.2 | 68 | 25.2 | 19 | 21.1 | |
| | Respondent and husband partner jointly | 132 | 36.7 | 117 | 43.3 | 15 | 16.7 | |
| Travel time to health facility | <60 minutes | 285 | 79.4 | 214 | 79.6 | 71 | 78.9 | 0.98 |
| | 60 to <120 minutes | 40 | 11.1 | 30 | 11.2 | 10 | 11.1 | |
| | ≥120 minutes | 34 | 9.5 | 25 | 9.3 | 9 | 10.0 | |
| | Missing | 1 | | 1 | | 0 | | |
| Cost of travel (round trip) | <25 Ethiopian Birr | 290 | 80.8 | 220 | 81.8 | 70 | 77.8 | 0.15 |
| | 25 to <50 Ethiopian Birr | 24 | 6.7 | 14 | 5.2 | 10 | 11.1 | |
| | ≥50 Ethiopian Birr | 45 | 12.5 | 35 | 13.0 | 10 | 11.1 | |
| | Missing | 1 | | 1 | | 0 | | |

‡ Chi-square test used for categorical variables; t-test used for continuous variables.

† Continuous variable, mean and standard deviation (SD) presented.

users of condoms with long-acting contraception (IUDs or implants) were 7.9%. The primary source of accessing contraception was health centers (51.1%). Additionally, 60.4% of the participants were informed of the side effects of the contraceptive method they were using by a health or family planning worker (see Table 3). Among women who were not using contraception, 64 (71.9%) had no intention of using contraception in the future (result not shown).

## Predictors of contraceptive practice among sexually active women living with HIV

Among the modifying/enabling factors, the prevalence of contraceptive use among women accessing health centers for ART was 1.25 times higher than women who accessed ART through hospitals (adjusted prevalence ratio [APR] = 1.25; 95%CI 1.10–1.42) (see Table 4). Compared to women who reported not having any children after HIV diagnosis, having 2 or more children (APR = 1.31; 95%CI 1.09–1.58) after HIV diagnosis was also associated with

**Table 2. HIV-related, reproductive-related and HBM factors among sexually active women living with HIV in western Ethiopia according to use of contraceptives, 2018 (n = 360).**

| Characteristics | Categories | All women (N = 360) | | Contraceptive users (N = 270) | | Non-users of contraceptives (N = 90) | | p‡ |
|---|---|---|---|---|---|---|---|---|
| | | n | % | n | % | N | % | |
| Time since HIV diagnosis | ≤ 5 years | 125 | 34.8 | 91 | 33.8 | 34 | 37.8 | 0.61 |
| | 5 to < 10years | 123 | 34.3 | 96 | 35.7 | 27 | 30.0 | |
| | ≥10 Years | 111 | 30.9 | 82 | 30.5 | 29 | 32.2 | |
| | Missing | 1 | | 1 | | 0 | | |
| Time since HIV diagnosis (in years)† | | 7.2(3.9) | | | | | | |
| Time on ART | ≤5 years | 157 | 43.9 | 120 | 44.8 | 37 | 41.1 | 0.30 |
| | 5 to < 10years | 126 | 35.2 | 97 | 36.2 | 29 | 32.2 | |
| | ≥10 Years | 75 | 21.0 | 51 | 19.0 | 24 | 26.7 | |
| | Missing | 2 | | 2 | | 0 | | |
| Time on ART (in years) † | | 6.2(3.6) | | | | | | |
| Health status since ART started | Poor/quite poor | 1 | 0.3 | 0 | 0.0 | 1 | 1.1 | 0.43†† |
| | Neither good nor poor | 9 | 2.5 | 7 | 2.6 | 2 | 2.2 | |
| | Quite good | 22 | 6.1 | 16 | 5.9 | 6 | 6.7 | |
| | Very good | 328 | 91.1 | 247 | 91.5 | 81 | 90.0 | |
| Recent CD4 count | < 350 cells/μL | 78 | 22.8 | 58 | 22.7 | 20 | 23.3 | 0.91 |
| | ≥350 cells/μL | 264 | 77.2 | 198 | 77.3 | 66 | 76.7 | |
| | Missing | 18 | | 14 | | 4 | | |
| CD4 (cells/μL) † | | 601.1 (298.6) | | | | | | |
| Partner tested for HIV | Yes | 282 | 79.2 | 221 | 82.8 | 61 | 68.5 | 0.016 |
| | No | 27 | 7.6 | 17 | 6.4 | 10 | 11.2 | |
| | Don't know | 47 | 13.2 | 29 | 10.9 | 18 | 20.2 | |
| | Missing | 4 | | 3 | | 1 | | |
| HIV status of the partner (n = 282) | Negative | 94 | 33.3 | 77 | 34.8 | 17 | 27.9 | 0.54 |
| | Positive | 185 | 65.6 | 142 | 64.3 | 43 | 70.5 | |
| | Don't Know | 3 | 1.1 | 2 | 0.9 | 1 | 1.6 | |
| Number of children living at home | No children | 52 | 14.5 | 23 | 8.6 | 29 | 32.2 | <0.001 |
| | 1–2 children | 210 | 58.5 | 164 | 61.0 | 46 | 51.1 | |
| | 3 or more children | 97 | 27.0 | 82 | 30.5 | 15 | 16.7 | |
| | Missing | 1 | | 1 | | 0 | | |
| Number of children not currently living with mother at home | No children | 274 | 77.2 | 207 | 77.5 | 67 | 76.1 | 0.69 |
| | 1 child | 33 | 9.3 | 26 | 9.7 | 7 | 8.0 | |
| | 2 or more | 48 | 13.5 | 34 | 12.7 | 14 | 15.9 | |
| | Missing | 5 | | 3 | | 2 | | |
| Intention of all births in the last five years (n = 152) | Intended | 86 | 56.6 | 82 | 57.3 | 4 | 44.4 | 0.46†† |
| | Mistimed | 27 | 17.8 | 24 | 16.8 | 3 | 33.3 | |
| | Unwanted | 39 | 25.7 | 37 | 25.9 | 2 | 22.2 | |
| HIV-status of the mother at conception (n = 152) | HIV-negative | 3 | 2.0 | 3 | 2.1 | 0 | 0.0 | 0.25†† |
| | HIV-positive | 129 | 84.9 | 123 | 86.0 | 6 | 66.7 | |
| | Unknown | 20 | 13.2 | 17 | 11.9 | 3 | 33.3 | |
| Children born after learning HIV status | No children | 174 | 48.5 | 105 | 39.0 | 69 | 76.7 | <0.001 |
| | 1 child | 123 | 34.3 | 105 | 39.0 | 18 | 20.0 | |
| | 2–3 children | 62 | 17.3 | 59 | 21.9 | 3 | 3.3 | |
| | Missing | 1 | | 1 | | 0 | | |

(*Continued*)

**Table 2.** (Continued)

| Characteristics | Categories | All women (N = 360) | | Contraceptive users (N = 270) | | Non-users of contraceptives (N = 90) | | p‡ |
|---|---|---|---|---|---|---|---|---|
| | | n | % | n | % | N | % | |
| Perceived susceptibility | Low | 130 | 36.1 | 71 | 26.3 | 59 | 65.6 | <0.001 |
| | Medium | 155 | 43.1 | 133 | 49.3 | 22 | 24.4 | |
| | High | 75 | 20.8 | 66 | 24.4 | 9 | 10.0 | |
| Perceived severity | Low | 119 | 33.8 | 99 | 37.6 | 20 | 22.5 | 0.031 |
| | Medium | 125 | 35.5 | 87 | 33.1 | 38 | 42.7 | |
| | High | 108 | 30.7 | 77 | 29.3 | 31 | 34.8 | |
| | Missing | 8 | | 7 | | 1 | | |
| Perceived benefits | Low | 122 | 34.2 | 75 | 28.0 | 47 | 52.8 | <0.001 |
| | Medium | 120 | 33.6 | 99 | 36.9 | 21 | 23.6 | |
| | High | 115 | 32.2 | 94 | 35.1 | 21 | 23.6 | |
| | Missing | 3 | | 2 | | 1 | | |
| Perceived barriers | Low | 99 | 33.3 | 82 | 37.3 | 17 | 22.1 | 0.042 |
| | Medium | 99 | 33.3 | 71 | 32.3 | 28 | 36.4 | |
| | High | 99 | 33.3 | 67 | 30.5 | 32 | 41.6 | |
| | Missing | 63 | | 50 | | 13 | | |
| Cues to action | Low | 129 | 36.4 | 103 | 38.9 | 26 | 29.2 | 0.12 |
| | Medium | 107 | 30.2 | 73 | 27.6 | 34 | 38.2 | |
| | High | 118 | 33.3 | 89 | 33.6 | 29 | 32.6 | |
| | Missing | 6 | | 5 | | 1 | | |

‡ Chi-square test used for categorical variables; t-test used for continuous variables.

† Continuous variable, mean and standard deviation presented.

†† Fisher exact test used.

increased risk of contraceptive use. The prevalence of contraceptive use among unmarried women was 0.69 times lower than their married counterparts (APR = 0.69; 95%CI 0.50–0.95).

After adjusting for health facility, marital status, decision-making regarding income, number of children living at home, perceived benefits, and children born after learning HIV status, women with both high (APR = 1.49; 95%CI 1.20–1.861) and medium (APR = 1.55; 95%CI 1.28–1.87) perceived susceptibilities were more likely to utilize contraception compared to those with low perceived susceptibility.

## Discussion

Using the HBM, this study examined contraceptive practices in Ethiopia among sexually active WLHIV aged 18–49 years who reported being fecund and not wanting children within two years. Three-quarters of this population were using some form of contraception, which serves as a crucial step in pursuing reproductive goals as well as supporting the PMTCT programs by preventing unintended pregnancies [44]. However, we found that a quarter of the participants had unmet contraceptive needs, which requires further intervention. It was also demonstrated that prevalence of contraceptive use was significantly lower among unmarried women compared to married women, which could put sexually active unmarried WLHIV at increased risk of unintended pregnancies. However, accessing health centers for ART as well as a higher number of children being born after HIV diagnosis were associated with increased risk of

**Table 3. Selected contraceptive indicators among sexually active women living with HIV in western Ethiopia, 2018 (N = 270).**

| Variables | Categories | Frequency | Percent |
|---|---|---|---|
| Contraceptive method used (multiple) | Condoms | 101 | 37.4 |
| | Pills | 8 | 3.0 |
| | Injectables | 121 | 44.8 |
| | IUDs | 20 | 7.4 |
| | Implants | 77 | 28.5 |
| | Sterilization | 3 | 1.1 |
| | Emergency contraception | 1 | 0.4 |
| Categories of contraceptive used (n = 266) | Condoms use only | 40 | 15.0 |
| | Short-acting only (pills, injectables) | 89 | 33.5 |
| | **Dual**: Condoms and short-acting (pills, injectables) | 40 | 15.0 |
| | Long-acting only (IUDs, implants) | 76 | 28.6 |
| | **Dual**: Condoms and long-acting (IUDs, implants) | 21 | 7.9 |
| Final decision on contraceptive selection | You alone | 109 | 40.4 |
| | Provider | 36 | 13.3 |
| | Partner | 20 | 7.4 |
| | You and provider | 39 | 14.4 |
| | You and partner | 48 | 17.8 |
| | You, partner and provider | 18 | 6.7 |
| Partner support towards contraceptive use | Supportive | 223 | 82.6 |
| | Indifferent | 34 | 12.6 |
| | Not supportive | 7 | 2.6 |
| | Does not know I am using it | 6 | 2.2 |
| The reason that the partner opposes contraceptive use (n = 7) | Wants to have more children | 5 | 71.4 |
| | Religion | 1 | 14.3 |
| | Harms my health | 1 | 14.3 |
| Time on contraception | 12 months or less | 74 | 27.6 |
| | >12 to 36 months | 124 | 46.3 |
| | >36 months | 70 | 26.1 |
| | Missing | 2 | |
| Source of contraception | Hospital | 70 | 25.9 |
| | Health center | 138 | 51.1 |
| | Health post | 10 | 3.7 |
| | Private-for-profit | 29 | 10.7 |
| | Non-government organization | 23 | 8.5 |
| Counselled on side effects | Yes | 163 | 60.4 |
| | No | 107 | 39.6 |
| If counselled, informed what to do concerning side effects (n = 163) | Yes | 157 | 96.3 |
| | No | 6 | 3.7 |
| Was informed on other contraceptive methods | Yes | 165 | 61.1 |
| | No | 105 | 38.9 |
| Using the method of your choice | Yes | 258 | 95.6 |
| | No | 12 | 4.4 |

contraceptive use. Interestingly, perceived susceptibilities regarding conception were significantly associated with increased risk of contraceptive use.

Despite the high rate of contraceptive use, the high prevalence of mistimed (17.8%) and unwanted births (25.7%) over the 5-year period, as well as the high unmet need for

**Table 4. Factors associated with contraceptive practice among sexually active women living with HIV in western Ethiopia, 2018 (n = 356).**

| Characteristics | Categories | Contraceptive use | | | | Unadjusted PR (95% CI) | Adjusted PR (95% CI) |
|---|---|---|---|---|---|---|---|
| | | Yes | % | No | % | | |
| **Modifying/enabling factors** | | | | | | | |
| Health facility | Hospitals | 142 | 52.6 | 74 | 82.2 | Ref | Ref |
| | Health centers | 128 | 47.4 | 16 | 17.8 | **1.35(1.17–1.57)** | **1.25(1.10–1.42)** |
| Marital status | Married | 239 | 88.5 | 52 | 57.8 | Ref | Ref |
| | Unmarried | 31 | 11.5 | 38 | 42.2 | **0.55(0.39–0.78)** | **0.69(0.50–0.95)** |
| Decision-making regarding income | Respondent only | 85 | 31.5 | 56 | 62.2 | Ref | Ref |
| | Husband/partner/family | 68 | 25.2 | 19 | 21.1 | **1.30(1.03–1.63)** | 0.94(0.77–1.14) |
| | Respondent and husband/ partner jointly | 117 | 43.3 | 15 | 16.7 | **1.47(1.21–1.78)** | 1.18(0.99–1.41) |
| Number of children living at home | No children | 23 | 8.6 | 29 | 32.2 | Ref | Ref |
| | 1–2 children | 164 | 61.0 | 46 | 51.1 | **1.77(1.17–2.67)** | 1.25(0.87–1.80) |
| | 3 or more children | 82 | 30.5 | 15 | 16.7 | **1.91(1.26–2.90)** | 1.28(0.89–1.85) |
| Children born after learning HIV status | No children | 105 | 39.0 | 69 | 76.7 | Ref | Ref |
| | 1 child | 105 | 39.0 | 18 | 20.0 | **1.41(1.18–1.70)** | 1.15(0.97–1.36) |
| | 2 or more children | 59 | 21.9 | 3 | 3.3 | **1.58(1.32–1.88)** | **1.31(1.09–1.58)** |
| **Perception** | | | | | | | |
| Perceived susceptibility | Low | 71 | 26.3 | 59 | 65.6 | Ref | Ref |
| | Medium | 133 | 49.3 | 22 | 24.4 | **1.57(1.26–1.96)** | **1.55(1.28–1.87)** |
| | High | 66 | 24.4 | 9 | 10.0 | **1.61(1.28–2.04)** | **1.49(1.20–1.86)** |
| Perceived benefits | Low | 75 | 28.0 | 47 | 52.8 | Ref | Ref |
| | Medium | 99 | 36.9 | 21 | 23.6 | **1.34(1.08–1.66)** | 1.16(0.97–1.39) |
| | High | 94 | 35.1 | 21 | 23.6 | **1.33(1.07–1.65)** | 1.11(0.91–1.35) |

Adjusted for for health facility, marital status, decision-making regarding income, number of children living at home, perceived benefits, children born after learning HIV status, and perceived susceptibilities.

contraception at the time of data collection (25.0%), raises concerns about efficacy, accessibility, and utilization of contraception among WLHIV [9]. A cross-sectional study at Saint Paul's Hospital Millennium Medical College, Addis Ababa, Ethiopia also showed similar magnitude of unmet needs for contraception among married WLHIV (25.1%) [45]. Two-thirds of WLHIV who had an unmet need for contraception had no intention of using contraception in our study, which was consistent with a finding from Uganda (77.6%) [46]. Improving awareness and counselling regarding effective contraception is thus important. Moreover, improving the quality of contraceptive programs [47] could have a high impact on addressing unmet needs for contraception and unintended pregnancy.

Importantly, different contraceptive methods are available in Ethiopia: pills, condoms, injectables, implants, IUDs, vasectomy and tubal ligation. Our finding that showed 37.4% of contraceptive users used condoms (with 15% using it as their sole method of contraception) is consistent with a prior study in Addis Ababa, Ethiopia [19]. The use of condoms offers HIV prevention. However, reliance on condoms as a single contraceptive choice is suboptimal for pregnancy prevention (at 18% failure rate under typical use conditions) [17, 48]. Therefore, ensuring highly effective contraceptive use while addressing concerns related to HIV is an excellent opportunity to meet the reproductive goals of WLHIV.

Despite dual methods providing the best protection against both pregnancy and sexually transmitted infections (STIs)/HIV [17], only about a quarter of participants were using dual methods (7.9% used condoms with long-acting contraception and 15% with short-acting contraception). This finding was lower than that reported in a study from South Africa (33%)

[49]. The high prevalence of STIs among WLHIV (16.3%) [50], as well as the many WLHIV who are unaware of the HIV status of their partner (7.6% reported their partner was not tested and 13.2% reported they did not know whether or not their partner had been tested) and the high rate of serodiscordance (HIV-negative partner) in our study creates additional health concerns for PLHIV. This reinforces the need for more support regarding dual use of contraception both in HIV clinics and family planning clinics.

Among hormonal contraception, injectables were the most commonly used method. This conforms to a prior study conducted in Addis Ababa, Ethiopia [19]. In our study, the second most used hormonal contraception was implants (28.5%). Injectables and implants were also the most accepted methods among the general population of Ethiopia [35] although these methods are thought to interact with some ART drugs. This raises concern around reduced effectiveness of injectables and implants methods [15, 51]. Compared to implants, injectables have a higher failure rate under typical use conditions [17]. Moreover, the adoption of IUDs was very low in our study, which might be because of misconceptions, such as negative perceptions by healthcare providers regarding its safety, negative perceptions towards IUD provision for nulliparous women and WLHIV, as well as low knowledge of IUDs [52]. Dispelling misinformation about IUDs and ensuring when WLHIV can use it safely would therefore be helpful. Importantly, women who wish to limit childbirth or want long-term protection should be supported by receiving appropriate counselling on the safety and efficacy of long-acting contraception of their choice. It is also crucial to meet contraceptive needs that align with their changing fertility intention.

Importantly, some modifying and enabling factors were associated with contraceptive use. Women accessing ART at health centers were more likely to use contraception compared to women accessing ART at hospitals. This is concerning given that more contraceptive choices are available in hospitals. In our study, the majority of contraceptive users reported that they obtained their contraceptive methods from health centers. Given health centers are available closer to the community (distance), WLHIV might be more likely to obtain contraception from the health centers as hospitals are usually further away [53]. Enhancing quality counselling is crucial given that only about half of contraceptive users reported being counselled on contraceptive side effects and on what to do if side effects develop. This echoes findings in Addis Ababa, Ethiopia [54]. Taken together, our study highlights the importance of enhancing the quality of contraceptive information, counselling and services in HIV clinics as well as family planning clinics at all levels of health facilities.

Our findings reveal significant differences in the prevalence of contraceptive use between married and unmarried sexually active WLHIV, which raises concerns about unmarried women's access to contraception. This finding is supported by a study in Ethiopia, which showed that unmarried women were also more likely to experience unintended pregnancies [22] and abortion [55] compared to married women. Enhancing access to and utilization of contraceptive information and confidentiality is important because unmarried women might face judgmental attitudes regarding their reproductive options [56]. Providing important support and contraceptive provisions that consider the circumstances of sexually active unmarried women could bridge this gap.

The prevalence of contraceptive use among WLHIV who had two or more children since HIV diagnosis was 1.31 times higher than WLHIV who never had a child after HIV diagnosis. Challenges during previous pregnancy such as HIV-related stigma from health professionals [57], distress and fear related to maternal and child health, personal shame associated with being pregnant as a WLHIV, and uncertainty about the future of the unborn baby [58], might be reasons not to consider another pregnancy. Essentially, these underscore the need for appropriate reproductive strategies, both conception strategies and contraception. Given

number of children living at home and age were not significantly associated with contraceptive use among sexually active WLHIV in our study, further investigation using a larger study may be able to better examine the impact of number of children and age on contraceptive practice.

Our study found that women's high, as well as medium perceived susceptibilities towards unintended pregnancy had a substantial impact on contraceptive use compared with low perceived susceptibility. The perceived susceptibilities assessed the perceived chance of pregnancy (unintended) if contraception is not used. Given understanding susceptibilities motivate women to practise contraception, healthcare providers should incorporate these perceptions to guide counselling and education while supporting safer conception for those who wish to have a child.

Despite only showing an association in an unadjusted model, the effect of perceived benefits cannot be excluded. Women with a high perception about the benefits of contraceptive use had an increased risk of contraceptive use compared to women with a low perceived benefits (in an unadjusted model). Essentially, contraceptive counselling regarding long-term protection of contraception, convenience of methods, effectiveness, as well as dual protection might increase adherence [59] and uptake of contraceptive use [60]. Improving cultural acceptability and community opinion towards contraception may also enhance contraceptive use [61]. Furthermore, developing contraceptive counselling guidelines and policies based on the HBM would be valuable in supporting WLHIV to achieve their reproductive goals. It is also essential to understand more about the acceptability of side effects and issues related to specific methods. Furthermore, the perceived benefits should be seen in relation to perceived barriers. The contraceptive initiation often requires targeted interventions of improving benefits while reducing barriers. Indeed, our current study adds evidence to previous studies [27–30] that show that the HBM can provide crucial insights into individual behaviours that improve contraceptive practice.

## Strengths and limitations

A major strength of the study was that the study applied the HBM and examined contraceptive use among all sexually active WLHIV not just those in union or married. However, this study must be considered in light of some limitations. First, unmarried women might be reluctant to report recent sexual activity in Ethiopia due to the sensitivity of the questions and social desirability bias. Therefore, a few sexually active unmarried WLHIV might have been excluded from the analysis. Second, all the data were self-reported, which is subject to recall bias. However, the data were collected using a standardized questionnaire by experienced female data collectors who had never worked at the selected HIV clinics to minimize biases. Third, we cannot infer associations described to causality because of the cross-sectional nature of the data.

## Conclusion

The majority of sexually active WLHIV in our study were using contraception, which gives insights into the role contraception plays in meeting family planning goals as well as in supporting PMTCT programs. Despite this, our findings also suggest the need for ongoing counselling and access to effective contraception given the rate of unmet needs for contraception and unintended pregnancies. Further interventions are required to address factors that impede the use of contraception when WLHIV do not want to conceive, particularly in terms of increasing the uptake of highly effective contraception because of lower efficacy of some of the methods (e.g., condom use only). The concepts within the HBM could shape contraceptive counselling for the identified 'at risk' women to achieve their reproductive goals as well as prevent mother to child transmission of HIV although this might require further investigation

using an interventional study. Our findings also suggest the HBM would be a valuable resource for healthcare providers, program planners and policymakers to develop guidelines and policies for contraceptive counselling and choices.

## Acknowledgments

We would like to thank study participants and data collectors for making this research successful. We would also like to thank Natalia Soeters for language proof.

## Author Contributions

**Conceptualization:** Tesfaye Regassa Feyissa, Melissa L. Harris, Deborah Loxton.

**Data curation:** Tesfaye Regassa Feyissa.

**Formal analysis:** Tesfaye Regassa Feyissa.

**Funding acquisition:** Tesfaye Regassa Feyissa, Melissa L. Harris, Deborah Loxton.

**Investigation:** Tesfaye Regassa Feyissa.

**Methodology:** Tesfaye Regassa Feyissa, Melissa L. Harris, Peta M. Forder, Deborah Loxton.

**Project administration:** Tesfaye Regassa Feyissa.

**Resources:** Tesfaye Regassa Feyissa, Melissa L. Harris, Deborah Loxton.

**Software:** Tesfaye Regassa Feyissa.

**Supervision:** Melissa L. Harris, Deborah Loxton.

**Validation:** Tesfaye Regassa Feyissa, Melissa L. Harris, Peta M. Forder, Deborah Loxton.

**Writing – original draft:** Tesfaye Regassa Feyissa.

**Writing – review & editing:** Tesfaye Regassa Feyissa, Melissa L. Harris, Peta M. Forder, Deborah Loxton.

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
