## [Decision Letter · Decision Letter 0]

30 Jan 2020

PONE-D-19-30290

Contraceptive use among sexually active women living with HIV in western Ethiopia

PLOS ONE

Dear Mr Feyissa,

Thank you for submitting your manuscript to PLOS ONE. After careful consideration, we feel that it has merit but does not fully meet PLOS ONE’s publication criteria as it currently stands. Therefore, we invite you to submit a revised version of the manuscript that addresses the points raised during the review process.

We would appreciate receiving your revised manuscript by Mar 15 2020 11:59PM. To enhance the reproducibility of your results, we recommend that if applicable you deposit your laboratory protocols in protocols.io, where a protocol can be assigned its own identifier (DOI) such that it can be cited independently in the future. For instructions see: http://journals.plos.org/plosone/s/submission-guidelines#loc-laboratory-protocols

We look forward to receiving your revised manuscript.

Kind regards,

Zelalem T. Haile, PhD

Academic Editor

PLOS ONE

Journal Requirements:

3. Please amend your current ethics statement to address the following concerns: Please explain why was written consent was not obtained, how you recorded/documented participant consent, and if the ethics committees/IRBs approved this consent procedure.

4. Please include additional information regarding the survey or questionnaire used in the study and ensure that you have provided sufficient details that others could replicate the analyses. For instance, if you developed a questionnaire as part of this study and it is not under a copyright more restrictive than CC-BY, please include a copy, in both the original language and English, as Supporting Information.  If the original language is written in non-Latin characters, for example Amharic, Chinese, or Korean, please use a file format that ensures these characters are visible.

5. Please state whether you validated the questionnaire prior to testing on study participants. Please provide details regarding the validation group within the methods section.

We would like to thank study participants and data collectors for making this research successful. TRF is supported by The University of Newcastle International Postgraduate Research Scholarship (UNIPRS) and The University of Newcastle Research Scholarship Central 50:50 (UNRSC 50:50). Dr Melissa Harris is supported by an Australian Research Council Discovery Early Career Researcher Award (DECRA). We would also like to thank Natalia Soeters for language edition.

This study was partially supported by the Hunter Medical Research Institute/Greaves Family Postgraduate Top-Up Scholarship (Grant number G1701582). Wollega University (first author's employer organization) facilitated data collection process. The funders had no role in study design, data collection and analysis, decision to publish, or preparation of the manuscript.

Additional Editor Comments:

Please refer to the comments from the reviewers below to revise and resubmit your manuscript.

Reviewers' comments:

Reviewer's Responses to Questions

**Comments to the Author**

1. Is the manuscript technically sound, and do the data support the conclusions?

Reviewer #1: Yes

Reviewer #2: Yes

2. Has the statistical analysis been performed appropriately and rigorously? 

Reviewer #1: No

Reviewer #2: Yes

3. Have the authors made all data underlying the findings in their manuscript fully available?

Reviewer #1: No

Reviewer #2: No

4. Is the manuscript presented in an intelligible fashion and written in standard English?

Reviewer #1: No

Reviewer #2: Yes

5. Review Comments to the Author

Reviewer #1: This paper describes a cross-sectional study examining contraceptive use among women living with HIV in Ethiopia. The authors sought to describe types of contraceptives used, prevalence of unmet need, and predictors of contraceptive use among a sexually active women who were not currently pregnant or planning to become pregnant. They also employed the Health Believe Model to assess beliefs about facilitators and barriers to contraceptive use and how this was associated with use. The authors found that contraceptive use was high, and identified a number of factors associated with contraceptive use, including marital status, number of children born since HIV diagnosis, and perceived susceptibility to unintended pregnancy. The authors conclude that their findings highlight the need for strengthening family planning services.

This research addresses an important topic, and while none of the findings are surprising the data collection appears to be well conducted and the findings can contribute to a body of literature assessing unmet need. However, as written, I have major concerns with this manuscript that I would recommend addressing before publication, as well as a number of minor comments.

Major concerns:

1. There were a number of grammatical errors, unclear sentences, and paragraphs that mixed concepts. The manuscript is also much lengthier than necessary, and does not focus very well on the topic of interest. I have noted some of these below, but it needs more editing than what is listed below. While I do think the science is sound, I would not recommend publication until there is thorough editing for grammar, sentence structure, and flow.

2. It is unclear why stepwise selection was used “to control for potential confounders” when this appears to be an exploratory analysis to identify factors independently associated with contraceptive use. Stepwise regression is often used when you have a specific exposure and outcome of interest, and you want to identify factors that may be confounding the relationship between those two variables. Here, you did not identify a specific exposure of interest, but still used this technique to identify a final model which you then presented as adjusted ORs, but then also presented those that didn’t make it into the model as unadjusted ORs. It wasn’t really addressed what it means for something to be in the final model or not. I believe a more appropriate approach is to present the findings of the multivariate model without stepwise selection. Those that are significant at your pre-specified alpha level are then the variables that are independently associated with contraceptive use when controlling for the other variables in your model. I think this analysis should either be changed, or better explained how you interpret your results.

3. I couldn’t really identify how the authors defined unmet need. This is simple to include and is necessary to compare to other studies that are assessing unmet need.

Minor concerns:

Introduction

1. This section in general is lengthier than it needs to be.

2. The first paragraph has a number of sentences with unclear wording or grammatical errors. This includes lines 48-49 “The 2015 WHO guideline avoids medical or clinical barriers”, line 50 “Along this improvement”, and line 55 “pregnancies has also benefits”.

3. Line 60-61 “Due to ART [16, 17], WLHIV need clear reproductive life plans with ready access to contraception” – What does this mean? Because they are living longer?

4. Line 71: I believe this should say “prevention of both pregnancy and HIV transmission”.

5. Line 71-74: Most of these things would also be a concern for women without HIV- should focus on transmission.

6. Line 88: should say “which include”

Methods

7. Lines 100-104: A lot of this detail, like exact population size and distance to the capital, is unnecessary

8. Lines 109: Should say “fecundity” or “being fecund”.

9. Line 111-112: You say an inclusion criterion includes having a demand of contraception. How is this defined? Did they report this, or was it determined based on meeting eligibility criteria for a-c?

10. For a number of variables, including length of time the women were using the method(s) without interruption and time on ART in years, you have some categories that are overlapping (e.g. >12-36, �36).

11. Lines 176-177: I think the caps here are unnecessary. You use italics for emphasis elsewhere; I think you could use that here as well.

12. Lines 199-206: I’m a little confused about this. Was this done for a previous analysis or specifically for this analysis? If done for a previous analysis, I would just state that summary scores were derived using PCA and the methods have been previously described, then cite where this has happened. If this was done specifically for this analysis, I would move it to the statistical analysis section.

13. Lines 211 and 214: p-value shouldn’t be capitalized

14. Statistical analysis section: Poisson with robust standard errors to get prevalence ratios performs better than logistic regression for ORs with cross-sectional data.

Results

15. Line 223: This is the first time unmet need is introduced and it was not defined.

16. Table 1: This is stylistic, but it is easier for the eye if the n and % are near each other (so n is right aligned and % is left aligned)

17. Table 1: The first foot note doesn’t seem to actually appear in the table? Also footnotes should read left to right, top to bottom, so the other footnote should be switched.

18. Line 239: Should be “reporting”

19. Line 248: You should restate what susceptibility and severity are referring to

20. Lines 270-282: Referring back to my comment above about your statistical approach, this way of reporting is a bit confusing. If the ones you report as unadjusted did not remain significant in the final model, this should be stated.

21. Lines 285-299: This final section is descriptive and would make more sense to be presented earlier.

Discussion

22. Lines 314-315: How does the prevalence of unmet need found in your study compare to that found in other studies?

23. Line 333-335 “There is considerable evidence that utilizing pre-exposure prophylaxis (although not currently available in Ethiopia) could address some of the gaps related to HIV transmission among non-condom users [40]” This does not seem particularly relevant considering your population is living with HIV. Since this paper is very lengthy it would be better to stick with things relevant to your findings.

24. Line 361: This is repetitive

25. Line 362-364: Wording unclear

26. Line 382-383 “were about fourteen times more likely to use contraception compared to women who never had a child after HIV diagnosis” You should acknowledge here that you had an extremely small sample size and wide confidence intervals for this finding.

27. Lines 383-388: It seems like this would be the expected outcome, as women with more children are probably less likely to want more children and are likely older. Did you consider this, or look at this at all in your data? Seems like a more simple explanation than the ones listed here.

28. Line 391: “This perception evaluated the chance of unintended pregnancies.” It’s not clear to me what this means.

29. Line 396: It’s unclear to me what “ruled out” means in this context.

30. Line 398: I think the “perceived susceptibility” here should be perceived benefit?

31. Lines 399-402: “Essentially, improved perceptions regarding contraception such as long-term protection, convenience of methods, effectiveness as well as dual protection for these women are critical factors identified by the HBM in our study and might increase uptake of contraception.” Did your findings really support this? How do you interpret the finding that most of your “HBM factors” did not appear in your final model?

Reviewer #2: Reviewer’s report

Title: Contraceptive use among sexually active women living with HIV in western Ethiopia

Version: 1

Date: 21 January 2020

Reviewer’s report:

Thank you for the opportunity to review the manuscript entitled “Contraceptive use among sexually active women living with HIV in western Ethiopia by Feyissa et al. on a very important subject. This is a well- written manuscript that is grounded in the Health Belief Model with clear objective and well written introduction. The methods are sound. The results are adequately presented and well synthesized against previous evidence and the model adapted as the conceptual framework. The authors also acknowledged the weaknesses/limitations in their study. Their conclusions are consistent with the evidence and arguments presented and addressed the main question posed. I strongly believe this manuscript makes a meaningful contribution to the field of public health and therefore, should be considered for publication pending editor's decision.

Nonetheless, these are few suggestions, which I think could strengthen the paper before publication.

Abstract

1. This is a well written abstract. However the authors can add a brief background to the abstract.

2. Line 21 Please specify the exact date (e.g 1st March to 1st June 2018)

3. Line 29 should read… were associated with increased “odds of” contraceptive practice.

4. Line 41, keywords: Health belief should be “health belief model”

Introduction

5. Very good introduction with adequate review of literature. However, this can be more strengthened when discussed within the context of the SDGs. Specifically goal 3.3

6. Line 63-64, can you please specify some of these studies?

Conceptual framework

7. Please this is a very minor comment. I would suggest the authors bring a sub-heading “conceptual framework” and also provide a diagrammatical representation of the variables considered in the model and how they are related to the outcome variable (s).

Materials and Methods

8. Line 98 should be read “Materials and Methods”

9. Is it possible to provide a map indicating the study settings?

10. Can you provide a sample of the instrument used for the data collection as an attachment (supplementary file)?

11. Line 101 can you please specify the exact date?

12. From Figure 1, the total women surveyed are 1082. Was this the total number of HIV positive women attending HIV clinics across the 4 facilities you were able to reach? If yes, how many were in each facility? How did you ensure representativeness across the facilities?

13. Although you have given a very nice flow of the way the 360 sample was arrived at but kindly explain how the systematic random sampling was used to arrive at the unit of analysis.

14. Line 119, please where was the piloting done? and what informed the choice of that place for the piloting? How many WLHIV were used for the piloting?

15. Line 122, please how long were the nurses trained and on what?

16. Line 169-170, please how was data on CD4 counts collected, self reported?? or from their records?? this should be acknowledged at the discussion section?

17. Line 203-205, please what informed the categorization of the “susceptibility” into tertiles (example: high susceptibility, medium susceptibility and low susceptibility?

18. Statistical analyses.

a. Please how was the adjustment done? (see Table 3).

b. Please what informed the choice of the reference categories?

c. Kindly specify how the missing values were treated at the inferential analysis stage.

Results

19. I suggest you bring the Table 4 (Selected contraceptive indicators among sexually active women with HIV in western Ethiopia, 2018) before Table 3 (Factors associated with contraceptive practice among sexually 283 active women living with HIV in western Ethiopia, 2018) to make all the descriptive results come before the inferential results(logistic regression).

Discussion

20. Line 304, the sentence… “supporting the PMTCT programs by preventing unintended pregnancies” should be supported with evidence

21. The authors have done an excellent job by acknowledging the weaknesses of their study including the study design, however, they did not acknowledge the strength of their study. I also suggest this should be given a sub-heading for easy reading.

Conclusion

22. Well written conclusions emanating from the study’s findings. Despite this can you please bring a subheading “conclusions and policy implications” directly after the strength and weaknesses of the study to make it easy for readers who wish to go to the conclusion section directly do that at ease.

---

## [Author Response · Author response to Decision Letter 0]

14 Mar 2020

Response to reviewers has been attached as a file.

---

## [Decision Letter · Decision Letter 1]

8 Jul 2020

PONE-D-19-30290R1

Contraceptive use among sexually active women living with HIV in western Ethiopia

PLOS ONE

Dear Dr. Feyissa,

Thank you for submitting your manuscript to PLOS ONE. After careful consideration, we feel that it has merit but does not fully meet PLOS ONE’s publication criteria as it currently stands. Therefore, we invite you to submit a revised version of the manuscript that addresses the points raised during the review process.

We look forward to receiving your revised manuscript.

Kind regards,

Zelalem T. Haile, PhD

Academic Editor

PLOS ONE

Journal Requirements:

Additional Editor Comments (if provided):

I am happy to inform you that it is provisionally accepted for publication pending your response to the minor points made by the review. Therefore, I invite you to respond to these comments and revise your manuscript accordingly. A rapid response on your part will facilitate a prompt publication process.

Reviewers' comments:

Reviewer's Responses to Questions

**Comments to the Author**

1. If the authors have adequately addressed your comments raised in a previous round of review and you feel that this manuscript is now acceptable for publication, you may indicate that here to bypass the “Comments to the Author” section, enter your conflict of interest statement in the “Confidential to Editor” section, and submit your "Accept" recommendation.

Reviewer #2: All comments have been addressed

Reviewer #3: All comments have been addressed

Reviewer #4: (No Response)

2. Is the manuscript technically sound, and do the data support the conclusions?

Reviewer #2: Yes

Reviewer #3: Yes

Reviewer #4: Yes

3. Has the statistical analysis been performed appropriately and rigorously? 

Reviewer #2: Yes

Reviewer #3: Yes

Reviewer #4: Yes

4. Have the authors made all data underlying the findings in their manuscript fully available?

Reviewer #2: No

Reviewer #3: Yes

Reviewer #4: Yes

5. Is the manuscript presented in an intelligible fashion and written in standard English?

Reviewer #2: Yes

Reviewer #3: Yes

Reviewer #4: Yes

6. Review Comments to the Author

Reviewer #2: The authors to a greater extent have addressed all my major concerns. Therefore, pending Editors decision, the manuscript should be accepted for publication.

Reviewer #3: Review of contraceptive use among sexually active women living with HIV in western Ethiopia

Thank you for the opportunity to review this paper which reviewed contraceptive use among sexually active women living with HIV. The researchers found that contraceptive use amongst sexually active WLHIV was found to be high compared to previous studies. WLHIV having two or more children after HIV diagnosis and with high and medium perceived susceptibility towards unintended pregnancy were more likely to use contraception. However, unmarried women were less likely to use contraception.

This paper is well written. Please find some minor editions from attached documents

Kind regards,

Reviewer #4: Remarks to the Authors

The manuscript entitled as “Contraceptive use among sexually active women living with HIV in western Ethiopia” is a research study that was conducted among sexually active women living with HIV in Ethiopia. This is an interesting study aimed at examining contraceptive practice among sexually active women living with HIV and identify the factors that influenced such practice using the Health Belief Model. Overall the study is clear and the results are consistent with the aim of the study. However, I recommend the following minor changes for the manuscript.

Introduction

1. The Introduction is well organized. However, the concepts described in the introduction are numerous, can be shortened for clarity.

2. Line 49- better to write it as ‘World health organization (WHO)’

Materials and methods

1. Line 102- Better to remove the sentence on line 102. ‘Nekemte and Gimbi are the capitals of the respective zones.’

References

1. The reference lists should be re-checked for clarity. E.g. Ref - 17, and 18.

2. Write in full words abbreviations/acronyms used in reference lists. E.g. Ref - 1, 2, and 3.

3. Write in small letter. Ref. 4.

---

## [Author Response · Author response to Decision Letter 1]

9 Jul 2020

Dear Dr Zelalem T. Haile

Thank you for considering our recent manuscript “Contraceptive use among sexually active women living with HIV in western Ethiopia”. We greatly appreciate the effort and comments from the reviewers. 

In response to your letter dated July 08, 2020 regarding our manuscript, we would like to provide the following clarifications and amendments. We have attached both a clean and annotated version of the revised manuscript.

Reviewer #2

The authors to a greater extent have addressed all my major concerns. 

Therefore, pending Editors decision, the manuscript should be accepted for publication.

Response: Thank you for your time in reviewing the previous and current versions of our manuscript. The comments were very helpful. 

Reviewer #3

Review of contraceptive use among sexually active women living with HIV in western Ethiopia. 

Thank you for the opportunity to review this paper which reviewed contraceptive use among sexually active women living with HIV. The researchers found that contraceptive use amongst sexually active WLHIV was found to be high compared to previous studies. WLHIV having two or more children after HIV diagnosis and with high and medium perceived susceptibility towards unintended pregnancy were more likely to use contraception. However, unmarried women were less likely to use contraception.

This paper is well written. Please find some minor editions from attached documents

Kind regards,

Response: We have amended the manuscript as per your editions. 

Reviewer #4

The manuscript entitled as “Contraceptive use among sexually active women living with HIV in western Ethiopia” is a research study that was conducted among sexually active women living with HIV in Ethiopia. This is an interesting study aimed at examining contraceptive practice among sexually active women living with HIV and identify the factors that influenced such practice using the Health Belief Model. Overall the study is clear and the results are consistent with the aim of the study. However, I recommend the following minor changes for the manuscript.

Introduction

1. The Introduction is well organized. However, the concepts described in the introduction are numerous, can be shortened for clarity.

Response: We have edited for clarity and reduced the length of the introduction section. 

2. Line 49- better to write it as ‘World health organization (WHO)’

Response: As per the comment, we have amended the statement and reads, ‘The 2015 World Health Organization (WHO) guideline states there should be no criterion barrier in the initiation of antiretroviral therapy (ART) [4] which will help in reductions in HIV-related morbidity and mortality [5]’. 

Materials and methods

1. Line 102- Better to remove the sentence on line 102. ‘Nekemte and Gimbi are the capitals of the respective zones.’

Response: We have deleted the statement.

References

1. The reference lists should be re-checked for clarity. E.g. Ref - 17, and 18.

2. Write in full words abbreviations/acronyms used in reference lists. E.g. Ref - 1, 2, and 3.

3. Write in small letter. Ref. 4.

Response: As per the suggestions, we have amended the references.

---

## [Editor Report · Decision Letter 2]

23 Jul 2020

Contraceptive use among sexually active women living with HIV in western Ethiopia

PONE-D-19-30290R2

Dear Dr. Feyissa,

We’re pleased to inform you that your manuscript has been judged scientifically suitable for publication and will be formally accepted for publication once it meets all outstanding technical requirements.

Kind regards,

Zelalem T. Haile, PhD

Academic Editor

PLOS ONE
---

## [Editor Report · Acceptance letter]

27 Jul 2020

PONE-D-19-30290R2 

Contraceptive use among sexually active women living with HIV in western Ethiopia 

Dear Dr. Feyissa:

I'm pleased to inform you that your manuscript has been deemed suitable for publication in PLOS ONE. Congratulations! Your manuscript is now with our production department. 

Kind regards, 

on behalf of

Dr. Zelalem T. Haile 

Academic Editor

PLOS ONE